# The Pivotal Role of Viruses in the Pathogeny of Chronic Lymphocytic Leukemia: Monoclonal (Type 1) IgG K Cryoglobulinemia and Chronic Lymphocytic Leukemia Diagnosis in the Course of a Human Metapneumovirus Infection

**DOI:** 10.3390/v13010115

**Published:** 2021-01-16

**Authors:** Jérémy Barben, Alain Putot, Anca-Maria Mihai, Jérémie Vovelle, Patrick Manckoundia

**Affiliations:** Geriatrics Internal Medicine Department, University Hospital of Dijon, Hôpital de Champmaillot 2, Rue Jules Violle—BP 87 909—21079 DIJON CEDEX, France; alain.putot@chu-dijon.fr (A.P.); anca-maria.mihai@chu-dijon.fr (A.-M.M.); jeremie.vovelle@chu-dijon.fr (J.V.); patrick.manckoundia@chu-dijon.fr (P.M.)

**Keywords:** aged, chronic lymphocytic leukemia, cryoglobulinemia, human metapneumovirus, monoclonal B-cell lymphocytosis

## Abstract

Background: Type-1 cryoglobulinemia (CG) is a rare disease associated with B-cell lymphoproliferative disorder. Some viral infections, such as Epstein–Barr Virus infections, are known to cause malignant lymphoproliferation, like certain B-cell lymphomas. However, their role in the pathogenesis of chronic lymphocytic leukemia (CLL) is still debatable. Here, we report a unique case of Type-1 CG associated to a CLL transformation diagnosed in the course of a human metapneumovirus (hMPV) infection. Case presentation: A 91-year-old man was initially hospitalized for delirium. In a context of febrile rhinorrhea, the diagnosis of hMPV infection was made by molecular assay (RT-PCR) on nasopharyngeal swab. Owing to hyperlymphocytosis that developed during the course of the infection and unexplained peripheral neuropathy, a type-1 IgG Kappa CG secondary to a CLL was diagnosed. The patient was not treated for the CLL because of Binet A stage classification and his poor physical condition. Conclusions: We report the unique observation in the literature of CLL transformation and hMPV infection. We provide a mini review on the pivotal role of viruses in CLL pathophysiology.

## 1. Introduction

Type-1 cryoglobulinemia (CG) is a rare disease associated with B-cell lymphoproliferative disorder, mainly nonmalignant monoclonal gammopathy of undetermined significance (MGUS). The association between Type-1 CG and Chronic Lymphocytic Leukemia (CLL) is rare [1]. A possible precursor of CLL is known as Monoclonal B-cell Lymphocytosis (MBL) [2]. Some viral infections, such as Epstein-Barr Virus, are known to cause malignant lymphoproliferation, like certain B-cell lymphomas. However, their role in the pathogenesis of CLL is controversial. Here we report the only case in literature of CLL diagnosis revealed by a human metapneumovirus (hMPV) infection, followed by a mini-review on the role of viruses in CLL pathophysiology.

## 2. Case Presentation

A 91-year-old man hospitalized in our geriatric center for delirium. His medical history consisted of a non-investigated major neurocognitive disorder and peripheral neuropathy of the lower limbs for 4 years with no etiological diagnosis. His usual treatment was ginkgo biloba extract and grape seed extract. He lived at home. His family reported an increase in behavior disturbances associated with cough over the 3 previous weeks.

At admission, the physical examination revealed fever, rhinorrhea, cough and widespread bronchi attributed to bronchitis. The neurological examination showed a bilateral loss of epicritic sensitivity in the lower limbs, but normal motor function. Delirium was also noted. There was isolated cervical lymphadenopathy. Initial biological tests showed normal levels of hemoglobin, platelets and leukocytes. A nasopharyngeal swab was taken on admission in order to perform molecular assay for respiratory viruses (i.e., multiplexed in-house RT-PCR for Influenza virus A and B, Rhinovirus/Enterovirus, human Metapneumovirus, Parainfluenza virus, adenovirus, non-SARS-Cov-2 coronavirus) identified a human metapneumovirus (hMPV) infection. Serology for hMPV was not performed.

The initial course was favorable with symptomatic care (aerosols of saline solution, paracetamol and respiratory physiotherapy) plus low-dose benzodiazepine for the delirium. Five days after viral diagnosis, the lymphocyte count increased from 1.16 to 10.86 × 10^9^ cells/L, while the symptoms of the infection remained under control. Serum protein electrophoresis suggested inflammation. The immunophenotyping of lymphocytes showed a profile of CD 5+, CD 23+, low FMC-7, and CD19b-, which is consistent with B-chronic lymphocytic leukemia.

We did not find other lymphadenopathies, splenomegaly or hepatomegaly during the physical examination or on imagery. He had no B symptoms other than fever consecutive to his infection (i.e., nighttime sweats and weight loss).

The lymphocyte count fluctuated considerably during hospitalization, with a maximum of 12.39 × 10^9^/L (Table 1). We did not perform FISH or IGHV mutation testing.

Because of the novel hematological finding and the unclear history of peripheral neuropathy, other laboratory tests, including for cryoglobulins, were performed. Renal and hepatic function were normal. The lab results found monoclonal IgG K cryoglobulinemia (0.1 g/L) associated with an increase in the activity of rheumatoid factor (7.5 IU/mL; normal < 3.5) and C3 (1.756 g/L; normal 0.811–1.570). There was a decrease in C4 activity (0.017 g/L; normal 0.129–0.392). Hepatitis C serology was negative, as were antinuclear antibodies.

The diagnosis of CLL associated with type-1 IgG K cryoglobulinemia revealed by hMPV infection was retained. The peripheral neuropathy was linked to an isolated clinical manifestation of cryoglobulinemia.

Given the absence of disability, significant gait disorders or skin lesions, the advanced neurocognitive condition, a Binet A (or Rai stage 1) CLL classification and the patient’s age, the care team decided not to treat the CLL. After discharge from the hospital, the patient was admitted to a nursing home.

## 3. Mini-Review and Discussion

### 3.1. Overview of Oncogenic Viruses

The link between viral infections and cancer is now well established, and about 15 to 20% of human cancers are thought to be viral in origin [3]. The main known oncogenic viruses are: Epstein-Barr Virus (EBV), involved in certain solid cancers and certain blood malignancies (see below) [4]; Hepatitis B virus (HBV), involved in hepatocellular carcinoma [5,6]; Hepatitis C virus (HCV) [7], the cause of certain forms of B-cell non-Hodgkin lymphoma [8]; human papillomavirus (HPV), involved in most cervical carcinomas and some head and neck cancers [9]; human T- lymphotropic virus 1 (HTLV-1), implicated in almost all T-cell leukemia [10]; Kaposi sarcoma-associated herpesvirus (KSHV, also known as human herpesvirus 8 (HHV-8)), involved in Kaposi’s sarcoma observed during deep lymphopenia in patients with AIDS-stage human immunodeficiency virus [11]; and Merkel cell polyomavirus (MCPyV), involved in about 80% of Merkel cell carcinoma cases [12].

The oncogenic potential of these viruses is the result of various strategies, both at the cellular and molecular levels, i.e., targeting tumor suppressor pathways or host signaling pathways, exploiting host DNA damage response, and manipulating the host immune response [13].

### 3.2. Viruses in Lymphoproliferative Disorders

In the case of lymphoproliferative disorders, the best known example of viral involvement is that of EBV [14]. First described by Epstein and Barr in the implication in Burkitt’s Lymphoma [15], EBV is actually involved in a broad range of B-cell lymphoproliferative disorders (i.e., classic Hodgkin’s lymphoma, lymphoma arising in immunocompromised individuals, HIV-associated lymphoma, aggressive lymphoma of the elderly), certain T-cell lymphoproliferative disorders (angioimmunoblastic T-cell lymphoma, extranodal nasal type natural killer/T-cell lymphoma and other rare subtypes) [16,17,18], undifferentiated nasopharyngeal carcinoma, and gastric cancer [19].

The mechanisms of viral-induced lymphoma are not well understood. One explanation could be the upregulation of the BcR (B cell Receptor), like in HCV-induced lymphomas [20]. Similar to HCV, HBV-induced diffuse large B-cell Lymphoma (DLBCL) has been epidemiologically demonstrated, but little is known about its pathophysiology. More than chronic antigenic stimulation, HBV-related lymphomagenesis could be explained by a wide range of specific genetic changes due to chronic HBV infection [21].

### 3.3. Pathophysiology of Chronic-Lymphocytic Leukemia and Virus Involvement

CLL represents 30 to 40% of newly diagnosed hematologic malignancies in adults in western countries, mostly in the elderly population [22]. The pathophysiology is complex, resulting from chromosomal and molecular alterations and numerous somatic mutations [23,24].

Although it is included in B-cell lymphoproliferative syndromes, CLL has a distinct pathogenesis. Often preceded by benign monoclonal B-cell lymphocytosis (MBL), defined as the presence of a clonal B-cell population in the peripheral blood with less than 5 × 10^9^/L B-cells with a CLL immunophenotype (mature CD5+ cells) and no other signs of a lymphoproliferative disorder [25], it then transforms into leukemic disease. In B-lymphoproliferative disorders, the BcR plays a key role in the development, proliferation, and survival of tumor B cells, by an antigen-driven process that triggers a molecular cascade of events that lead to transcriptional activation of proinflammatory, proliferative and antiapoptotic genes [26,27]. While certain signaling pathways involved in EBV-induced lymphomas are common in CLL, such as the Notch signaling pathway [28], the virus-induced nature of CLL has never been demonstrated [24]. CLL could be subdivided into two main subsets that depend on the mutation status of the immunoglobulin heavy-chain variable region gene (IGHV) of BcR, reflecting the origin of the malignant cells [29]. The origin of the more aggressive disease with unmutated IGHV (U-IGHV) is taken from a B-cell that has not undergone differentiation in a germinal center (the place in lymph nodes were T and B lymphocytes interact to permit somatic hypermutations (SHM) in immunoglobulin variable region genes of B-cell and selection during immune response) [30]. Mutated IGHV (M-IGHV) is the second main subset of CLL cells.

Similar to what happens during an immune response to an antigen, these cells arise from a post-germinal center and express immunoglobulin that has undergone SHM [30,31]. The prognosis of M-IGHV CLL is more favorable [32]. Since the immunoglobulin variable regions are B cell specific, the sequence of BCR allows the identification of single B-cell clones. Of note, it was well-documented the co-existence of different CLL clones in a single patient [33]. CLL cells are mainly characterized by a significant restriction of the repertoire of immunoglobulins, reflecting the biased use of certain IGHV genes in CLL. BcR stereotypy is found in almost one-third of CLL-patients [34,35].

It is supposed that BcR immunoglobulins belonging to the same stereotyped subset could be selected by a restricted range of antigenic epitopes [31,36]. In fact, the role of viruses in CLL pathogenesis appears more conceivable (Figure 1). HCV was particularly relevant, with the stereotyped IGHV4-59/IGKV3-20 BcR subset in CLL-patients. However, the direct or indirect link with HCV infection could not be demonstrated [37]. EBV was also associated with CLL, notably with a poor prognosis, but the role in the pathogenesis could not be established, in contrast to B-cell lymphomas [38].

Some studies have argued in favor or a cross-reaction between U-CLL cell BcR and HIV and HCV [39].

### 3.4. Prognostic Role of Viruses in CLL

One of the most serious complications of CLL is a transformation into DLBCL, also called Richter’s syndrome (RS). EBV infection is clearly involved in that progression, but HBV or HCV infection after CLL diagnosis do not seem have a prognostic effect [40,41,42]. However, one Chinese study proposed a clinical model in which HBV infection status was associated with time-to-treatment and overall survival in CLL [43].

Another case in which a virus can be considered to have a prognostic role is in Merkel cell carcinoma (MCC) caused by MCPyV since it is more frequent in CLL patients [44]. On the contrary, some studies suggest that there is a higher risk of CLL and non-Hodgkin lymphomas in MCC patients [45]. However, despite a lymphotropism for MCPyV and detection of the virus in CLL-cell patients, there is currently no evidence of MCPyV involvement in CLL pathogenesis [46,47,48].

### 3.5. hMPV Involvement in B-Cell Malignancies?

hMPV is a recently-discovered (2001) single-strand RNA paramyxovirus that is frequently involved in respiratory tract infections in infants and elderly people, ranging from mild upper respiratory tract disease to severe bronchiolitis and pneumonia [49]. The development of T and B cell memory is poor and may therefore promote reinfection [50]. While infection with this type of virus is an undeniable clinical challenge in individuals who are immunocompromised by a neoplastic disease, its tumorigenic role has never been demonstrated [51].

## 4. Conclusions

Although only a temporal link can be made, our case clearly raises the question of whether hMPV had a role in promoting CLL-like lymphocytosis in a patient with anterior MBL. Viruses appear to be pivotal for the pathophysiology CLL, as well as other B-cell malignancies.

In summary, we report the case of CLL associated with a type-1 CG diagnosis made during the course of an hMPV infection. While the association between the hematologic disease and the viral infection could not be affirmed, this case raises questions about the viral promotion of CLL transformation.

## Figures and Tables

**Figure 1 viruses-13-00115-f001:**
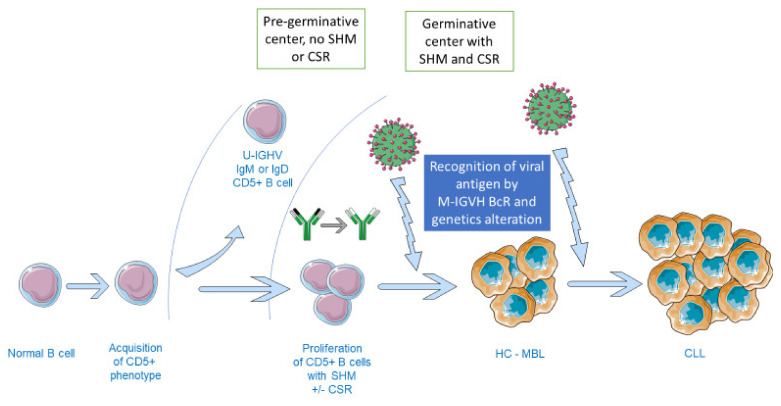
Hypothesis for virus involvement in CLL pathophysiology. Some subsets of B cells acquire CD5+ expression. Without SHM (high-frequency point mutations in Ig heavy and light chains that occur in germinal center B cells in response to signals from T follicular helper cells. Mutations that result in increased affinity of antibodies for antigen impart a selective survival advantage to the B cells producing those antibodies and lead to affinity maturation of the humoral immune response), a subset of these CD5+ B cells which do not penetrate the GC could lead to unmutated IGHV CLL, a condition with a poor prognosis. For the other subset of CD5+ B cells, they penetrate the GC where SHM occurs with IGHV mutations and, for certain, CSR. The recognition of a viral antigen by the M-IGHV BcR Ig of CD5+ cells may occur some immunogenetic changes in the cell and acquire pathological clonal expansion called MBL. After some additional immunogenetic changes, like after exposure to a new virus, the most severe form of MBL, HC-MBL (LC-MBL is not represented), could transform into CLL. BcR: B cell Receptor, CLL: Chronic lymphocytic leukemia, CSR: Class switch recombination, GC: germinal center, HC or LC—MBL: High-count or Low-count monoclonal B-cell lymphocytosis, M/U IGHV: mutated/unmutated immunoglobulin heavy-chain variable region gene, SHM: Somatic hypermutation. Note that the virus is not physically present in the GC.

**Table 1 viruses-13-00115-t001:** Variation in lymphocyte and platelet counts and hemoglobin level. hMPV: human Meta Pneumovirus. CLL: chronic lymphocytic Leukemia.

	2015—Peripheral Neuropathy Diagnosis	2019—Hospitalization Day 0. hMPV Infection Diagnosis	2019—Hospitalisation Day 5. CLL Diagnosis	2019—Hospitalization Day 13	2019—Hospitalization Day 15. Type-1 IgG K Cryoglobulin Diagnosis	2019—Hospitalization Day 41. Discharge	2019—4 Months after Discharge
Lymphocyte count (×10^9^/L)	0.87	1.16	10.86	12.39	5.8	7.36	11.1
Platelet count (×10^9^/L)	180	189	386	467	334	237	412
Hemoglobin level (g/dL)	13.5	13.9	13.9	13.6	13	12	12.4

## Data Availability

The data presented in this study are available on request from the corresponding author. The data are not publicly available due to the participant’s request.

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
