# Peer review of "The Pivotal Role of Viruses in the Pathogeny of Chronic Lymphocytic Leukemia: Monoclonal (Type 1) IgG K Cryoglobulinemia and Chronic Lymphocytic Leukemia Diagnosis in the Course of a Human Metapneumovirus Infection"

_viruses, 2021, doi:10.3390/v13010115_

Round 1

Reviewer 1 Report

The authors detected human metapneumovirus  (hMPV) in a patient that arrived to the hospital and was diagnosed with Type-1 cryoglobulinemia (CG) and Chronic Lymphocytic Leukemia (CLL).   hMPV is the second most common cause after Respiratory syncytial virus (RSV) of lower respiratory infection in young children. It may cause disease in all ages, and almost all children worldwide become seropositive by the age of five year. Importantly, this virus does not establish latency, and cleared from the patient within few weeks.  On the contrary, all known seven viruses that cause cancer in humans lead to chronic infection, and cancer development takes years. Another criterion to determine that a virus causes cancer is that all tumor cells are infected with the virus. It seems that the observation of hMPV in a patient with CLL does not stand in these two criterions: hMPV does not chronically persists and the authors have not presented any proof that all CLL cells are infected with hMPV. Based on these limitations we cannot accept the major findings of the manuscript

Author Response

We thank the reviewer for his or her comments.

Reviewer 2 Report

This manuscript describes onset of CLL in concomitance with a human Metapneumovirus infection; type 1 cryoglobulinemia was also diagnosed in this 91-year-old patient, but its concomitance with infection is made unclear by the pre-existence of peripheral neuropathy that could be related to already present cryoglobulinemia.

The paper is well-written and interesting, and the review on virus-related neoplastic disorders in the discussion is satisfactorily comprehensive, and the paper could be accepted although I suggest few minor changes.

Line 107. The Authors correctly state that the relationship between HBV and diffuse large B cell lymphoma has been convincingly demonstrated although its pathophysiology is poorly known, but at this regard they refer (ref. 21) to a 2015 study from China that claims hepatitis B surface antigen-driven origin. Indeed, the latter statement has been subsequently disconfirmed by a large study, also in Chinese patients, which provides a broad genetic landscape of these lymphomas. I suggest substituting ref. 21 with the latter paper by Ren et al.* (see also the related commentary in the same issue).

*Ren W, Ye X, Su H, et al. Genetic landscape of hepatitis B virus–associated diffuse large B-cell lymphoma. Blood. 2018;131(24):2670-2681.

Line 173. “hMPV had a role in promoting transient CLL-like lymphocytosis”. Indeed, the data provided in Table I do not support transiency of lymphocytosis, that persisted up to the last follow-up at month 4 after discharge; if Authors have evidence for a subsequent decline of lymphocytosis this should be stated, otherwise “transient” should be eliminated from the statement.

Reviewer 3 Report

The manuscript is clearly written. It will be nice to specify: 

  • line 117: …leukemic disease.

In B-lymphoproliferative disorders, the B cell receptor (BCR) plays a key role in the development, proliferation, and survival of tumor B cells [4], by an antigen-driven process that triggers a molecular cascade of events that lead to transcriptional activation of proinflammatory, proliferative and antiapoptotic genes [PMID: 30887112 - PMID: 30975981]. 

  • line 128: …more favorable (30).

Since the immunoglobulin variable regions are B cell specific, the sequence of BCR allows the identification of single B-cell clones. Of note, it was well-documented the co-existence of different CLL clones in a single patient ( PMID: 27568521 )

  • line 128: CLL cells are MAINLY characterized

  • line 130: add most recent reference PMID: 30779034 

Author Response

This manuscript is a resubmission of an earlier submission. The following is a list of the peer review reports and author responses from that submission.